# The impacts of project uncertainty and relationship conflict on relationship continuity in construction: The role of political skill

Xiaoyan Huo[1,2]*, Shiya Gao[1,2], Huiyang Zhang[1,2]

1 School of Urban Geology and Engineering, Hebei GEO University, Hebei, China, 2 Hebei Technology Innovation Center for Intelligent, Development and Control of Underground Built Environment, Beijing, China

* huoxiaoyan85021@163.com

## Abstract

Project uncertainty is a common feature in construction projects, and relationship conflicts often arise from interactions among project partners. This study examines the negative effects of project uncertainty and relationship conflict on relationship continuity, which is a key factor influencing collaboration and long-term mutual benefits in construction projects. While previous research has sufficiently addressed the role of formal mechanisms in promoting relationship continuity, this article emphasizes the importance of political skill as an effective soft mechanism. A moderated mediation model is proposed to investigate how political skill mitigates the negative impacts of project uncertainty and relationship conflict on relationship continuity. Data were collected through a structured questionnaire survey comprising 230 valid responses from 50 construction project teams, and hypotheses were tested using bootstrapping procedures. The results show that both project uncertainty and relationship conflict are negatively associated with relationship continuity. Furthermore, relationship conflict mediates the link between project uncertainty and relationship continuity, and this mediation effect is weakened when project partners demonstrate greater political skill. These findings underscore the value of political skill as a relationship management tool that can enhance the competitiveness of construction enterprises.

## Introduction

The effective management of organizational relationships is essential for the success of construction projects [1,2]. However, the temporary and dynamic nature of construction projects, coupled with uncertain environments, often leads to fragmented relationships and increased conflict, thereby emphasizing the critical need

**Data availability statement:** All relevant data are within the manuscript and its Supporting Information files.

**Funding:** This study was supported by National pre-research project of Hebei GEO University (KY2024YB18); Hebei Technology Innovation Center for Intelligent Development, Control of Underground Built Environment; Demonstration Undergraduate Majors for Applied-Oriented Transformation in Hebei Province.

**Competing interests:** The authors have declared that no competing interests exist.

for relationship continuity in this context [3]. Relationship continuity, which represents high-involvement and long-term collaborative partnerships, has received considerable attention in construction project management. It is primarily pursued to improve resource utilization and establish relationships characterized by commitment and trust among partners [3].

Nevertheless, project uncertainty is a major factor influencing relationship continuity [4–7]. Construction projects are characterized as uncertain and are always unpredictable; the activities involved in such projects cannot be anticipated during the early stages of the project [4]. Although project managers make great efforts to construct detailed plans and address various uncertainties, well-designed plans usually perform in unexpected ways. Uncertain problems that emerge throughout a project led to various conflicts among partners. In general, conflicts can be divided into two categories: task conflicts and relationship conflicts [8,9]. The former type of conflict manifests as differences in opinions, decisions, and ideas; and some scholars have found that this type of conflict can promote innovation and team performance, and is closely related to decision quality. Therefore, it can have a positive impact [8–10]. The latter type of conflict can easily lead to interpersonal incompatibility and hostility among team members, which is detrimental to maintaining organizational relationships, causing a decline in business performance and hindering high-quality decision-making. Therefore, it will have a negative impact on team performance, and in the actual process of enterprise management, relevant methods should be adopted to eliminate such conflicts [11]. Researchers have indicated that relationship conflicts leads to negative feedback, like animosity and disgust. As such conflict increases, partners no longer intend to engage in continuous partner relationships. As indicated, the project management literature features an important gap with repect to the mechanisms by which project uncertainty influences partners' relationship continuity through relationship conflict in the construction context.

Investigations in this field have increasingly focused on partners' relationship continuity [12–14]. According to one stream of research, project managers develop a series of formal governance mechanisms for establishing a collaborative relationship atmosphere [15–17]. Despite the successes of such formal mechanisms, many case studies have indicated that although a series of formal arrangements have been adopted, project partners have also encountered problems, such as a lack of the mindset necessary to establish a continuous relationship culture [18–20]. Bresnen and Marshall noted that current practice activity places excessive emphasis on formal mechanisms [18]. Project organization involves diverse individuals, and the dynamics underlying the relationships among partners are normal [19,21]. Cooke-Davies reported that individuals perform important functions in project activities and outcomes [22]. Nevertheless, the more distinctive of an individual is, the higher probability of they have interpersonal incompatibility, which affects the long-term cooperative relations. Thus, research on the essence of relationship continuity between partners is needed. Another research stream has looked beyond 'hard' operational processes (such as formal partnering arrangements) to focus more on 'soft' factors such as partners' relationship skills [16,21,23,24]. The construction project environment is

often akin to a political arena, with multiple participants pursuing their own interests, demands, and goals. These stakeholders frequently achieve win-win outcomes through alliance building, underscoring that partner capabilities are crucial for project success amidst increasing complexity [25–27]. Cousins et al. reported that project participants' relational skills have become one of the most critical factors regarding in project activities [28]. Unsurprisingly, calls for more systematic explorations of project partners' relationship skills are an essential part of construction research [25,26].

In line with the preceding discussion, many studies have indicated that political skill is a key relationship skill for project partners. Project organization is a diverse and dynamic type of organization that has been described as an inherently political context [29]. It is essential for partners to develop political skills that can help them to deal with relationship conflicts and establish continuous relationships with their partners most effectively. Ferris et al. described political skill as "the ability to effectively understand others at work and to use such knowledge to influence others to act in ways that enhance one's personal and/or organizational objectives" [30]. Numerous studies have reported that political skill is positively associated with job satisfaction,organizational citizenship behavior (OCB), and self-esteem [31–35]. It is imperative to acknowledge its dual nature. Political skill can be a double-edged sword; if wielded unethically, it can lead to manipulation and self-serving behavior. However, this study focuses on its constructive application in mitigating negative relational dynamics. Extensive research has demonstrated that political skill can buffer negative relationships, such as burnout-abusive supervision [36], abusive behavior-interpersonal conflict [37], and counterproductive work behavior-workplace ostracism relationships [38]. The findings of the present study reveal that political skill, can directly impact outcomes and predictor–outcome relationships [39]. This factor exhibits cognitive and behavioral manifestations and has been studied extensively in the fields of management and psychology. Additionally, based on the contingency theory, there is no universally applicable optimal management method for organizations, the effectiveness of a management approach depends on the degree of alignment between management practices and the external environment (such as uncertainty). Guided by this principle, this research focuses on political skill as a contingency factor to examine the ways in which it mitigats the negative impact of project uncertainty on partner relationship continuity.

To accomplish this goal, the present study aims to provide construction managers with substantial instructions pertaining to partner relationship continuity. Two critical objectives of this research are as follows. (1) Based on an analysis of the relationships between project uncertainty and relationship conflict and between relationship conflict and relationship continuity, we examine in further detail how project uncertainty influences relationship continuity through relationship conflict in the context of construction projects. (2) The present research applied contingency approach to explore the moderating role of political skill in project uncertainty on relationship continuity (through relationship conflict). Findings pertaining to these two objectives are discussed in further detail to explore partners' political skills and relationship management, which are closely associated with the relationship continuity of construction project partners. The overall framework for this research is presented in Fig 1.

## Literature review and hypothesis development

**Project uncertainty, relationship conflict and relationship continuity.** Construction projects are intrinsically uncertain. Since a project is a complex event that faces limitations in terms of time, costs, and resources, planning seems to be a difficult task, and uncertainty is a normal phenomenon in this context. The traditional project implementation

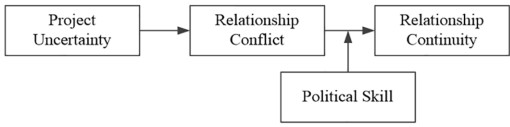

**Fig 1. Overall research framework.**

method follows linear logic such that the project can remain "on track" in accordance with a predetermined operating plan [40]. However, while project activities need to be planned in a timely manner, this factor cannot directly lead to project success.

Project uncertainty is rooted in certain events or situations, regardless of whether they can be considered in advance [5]. According to many literature reviews, researchers have defined project uncertainty main in terms of three aspects: project scope, technical complexity and project novelty [15,41]. The project scope refers to the project's cost and duration as well as the number of partners involved. Theoretically, the larger the scope of the project is, the higher the uncertainty level and the greater the amount of information needed by the project manager to coordinate and monitor the project's costs and schedule [42]. Technological complexity is largely the result of the innovativeness and complexity of the technology used in the project. In this case, the project team may not have a clear procedure for implementing a complex technology in the project. Therefore, partners must engage in rich communication and jointly address various problems and solutions to ensure that the goals of the project are met [43]. The novelty of the project pertains to the function of its organization and the novelty of project activities for the project partners. The more novel the project is, the more unpredictable the events and problems that the partners will encounter during the project are [44]. According to transaction cost theory [45], due to the uncertainty and various changes in environmental factors, both sides of the transaction will incorporate future uncertainty into the contract, which increases the transaction cost and leads to opportunistic behavior, thus affecting project performance. It is obvious that communication and coordination among all parties are essential for ensuring project uncertainty [46]. In addition, partners must interact at a higher level to discuss different points and views with the goal of reaching a consensus to ensure relationship continuity.

However, a construction project involves owners, contractors, and suppliers [47]. These individuals all have different backgrounds, technical skills, and personalities. Therefore, it is impossible to compensate for the uncertainties encountered in such a project simply by collecting additional information and performing comparisons. Partners' different ideas, values, and beliefs trigger relationship conflicts, which in turn lead to the deterioration of interorganizational relationships. First, according to information processing theory, the generally accepted definition of uncertainty focuses on the lack of information necessary to complete a given task [46]. This definition suggests that project partners should play an active role in facilitating problem discussions and integrating all viewpoints to reach a consensus. Second, self-categorization theory refers to the fact that individuals usually categorize themselves or others into different social groups and define groups according to the characteristics of individual diversity (superficial or deep), social structure, and work unit. When an individual identifies himself with a group that is similar to him and considers the individuals in the group to be similar to him. Individuals in such groups are more willing to exchange information and develop common perceptions and values. In this way, individuals in the group also tend to share information and communicate, thus stimulating further deep communication motivation and other interactive behaviors. Nevertheless, in response to unpleasant or unexpected events, people tend to attribute the behavior of others to their character to an excessive degree. When this attribution process focuses on personal attacks [48] or hidden agendas [49], such as harsh, aggressive behavior or unexpected events (uncertainty), it triggers relationship conflict through biased information processing [50,51]. Thus, the following hypothesis is proposed:

**H1.** Project uncertainty is positively associated with relationship conflict.

Organizational researchers have defined relationship conflict as referring to disagreements among team members regarding issues that are unrelated to the task at hand (e.g., value or beliefs). The literature has confirmed the dysfunctional effects of interpersonal conflict on team member satisfaction [8,10,52–55].

Scholars have also noted that when relationship conflicts occur in teams, the parties involved in the conflict often experience adverse emotions, such as irritability and anger. Kiefer noted that adverse emotions have a negative influence on behaviors such as individual effort [56]. In construction projects, the characteristics of boundary spanners in partnerships include cognitive complexity and behavioral complexity [21]. According to the theories of social identity and self-categorization, learning is regarded as a process of knowledge acquisition through information cognitive processes, and

the differences in individual information cognition are determined by individual characteristic differences. Individual psychological cognition achieves the expected results desired by the individual through the control behavior of the environment. This theory points out that the three factors of individual behavior, subject cognition, and the social environment are dynamically and interactively influence each other. Its core idea is that there is a causal relationship between human cognition and behavior, and at the same time, internal thinking activities and environmental factors jointly determine human behavior. Thus, partners are urged to maintain their self- and social identities, and they show prejudices that bias them in favor of people who appear to exhibit similar characteristics [57]. Specifically, such partners believe that their values and beliefs will be strengthened in such a context. When various values, personalities and beliefs are incompatible, relationship conflicts occur, which may trigger negative feedback (i.e., negative emotions), such as disgust. As adverse emotions increase, strong negative emotions are exacerbated, and partners with adverse emotions are more likely to focus on the task of pursuing long-term organizational relationships [58]. Hence, it can be concluded that relationship conflict is detrimental to relationship continuity.

**H2.** Relationship conflict is negatively associated with relationship continuity.

Davies noted that individuals are a key factor in project activities (such as communication) and project results [22]. Unforeseen events continually occur in the context of construction projects. Project partners must engage in rich communication and continuous interaction to resolve various uncertainties and problems to ensure project continuity. As indicated above, the project uncertainty of this study is mainly in terms of three aspects: project scope, technical complexity and project novelty. In response to unclear project scopes (e.g., in terms of duration, cost, or investment), partners must reach an agreement as soon as possible by employing various communication methods (e.g., negotiation, formal or informal meetings) to ensure project continuity. Moreover, construction projects are characterized by complex and novel technologies. To determine the appropriate method, partners must form a consensus and develop clear procedures. However, as long as interaction and interdependence between project partners continue to characterize this relationship, conflicts are common [59]. When the presence of different viewpoints causes a conflict, opinions lead to interpersonal attacks (i.e., abuse, harsh), and partners may experience feelings of hatred and disgust and engage in corresponding behaviors, which are sources of relationship conflict [10,53]. In addition, the relationship conflict experienced by a person is reflected in that person's negatively motivated behaviors, which can lead to defensive attitudes and reduced effort at work, although it is possible to establish a better cooperative relationship. Therefore, project uncertainty influences relationship continuity, and relationship conflict is a key mediating factor in this context. Hence, the following assumption is proposed:

**H3.** Relationship conflict mediates the association between project uncertainty and relationship continuity.

## The moderating role of political skill

Construction projects are characterized by uncertainty due to their dynamic and complex characteristics. Project partners should establish long-term and high-involvement relationships across projects. Nevertheless, as long as interactions between project partners occur, relationship conflicts are inevitable. The contingency theory of conflict management proposes reducing the negative impact of relationship conflict through reasonable management behavior and organizational design. Conflict managers should use contingency theory for people, for things and for times, and fully consider management background and contingency factors. In other words, it pays attention to the moderating variables between relationship conflict and relationship continuity, establishes a contingency model to analyze and explore the reasons behind the impact of relationship conflict on relationship continuity, and provides theoretical and practical guidance for managers in relationship conflict management.

Recent organizational research has revealed that political skills are important personal skills in the modern complex organizational environment [60,61]. As defined, political skill represents a person's ability to read and understand others and situations to take action on the basis of this knowledge to influence others, thereby achieving goals [34]. Ferris et al. proposed a common framework for political skills [34]. These authors proposed four types of political skills: social

insight, interpersonal leadership, social aptitude, and sincerity. The former two political skills of an individual refer to his or her keen understanding that a partner with these skills can read people accurately and take appropriate actions to elicit responses from others [34]. Social aptitude ability refers to the ability of individuals to establish alliances, use alliances, and engage in complex network relationships [34]. Hence, sincere people exhibit a high degree of sincerity, which helps them obtain confidence of the people they interacted [34].

Recent studies have shown that political skills have positive effects on the self (such as motivation and humility), others (such as accountability and trust), and groups/organizations (such as extended goals, influence/learning, and empowerment) and subsequently promote the achievement of performance results [62]. People with proactive personalities are likely to take proactive actions to impact their environments [63]. For example, scholars have reported that political skills are important predictors of job satisfaction, career success, personal reputation, OCB, and self-efficacy [31–35]. Moreover, political skill can be used to moderate the burnout-abusive management relationships [36], the abusive behavior-interpersonal conflict relationships [37], and the counterproductive work behavior-workplace ostracism relationships [38]. According to previous studies, political skills are comprehensive social competences that can take both cognitive and behavioral forms [64–67]. These factors not only impact the results directly but also have a moderating effect on predictor-outcome relationships.

Accordingly, abundant evidence has shown that that political skill moderates the influence of relationship conflict on relationship continuity. Moreover, political skill moderates the indirect impact of project uncertainty on relationship continuity (through relationship conflict). Specifically, the link between relationship conflict and partner relationship continuity is weaker for partners who exhibit higher political skill then for those who exhibit lower levels of such skill. Additionally, it moderates the indirect effect of project uncertainty on relationship continuity (through relationship conflict). Specifically, this indirect effect is weaker for partners with higher political skill than for those with lower political skill. Partners with higher political skills can understand situations and social interactions, sincerely use their ability to influence people effectively, and obtain the desired results from their interactions with others [39]. Project partners with political skills can mitigate the dysfunctional effects of relationship conflicts by reducing negative emotions (such as anxiety and frustration) at work [67]. Conversely, project partners with low levels of political skills may fail to counter the negative emotions caused by relationship conflict [68,69]. This ability does not involve denying one's negative emotions completely but rather focuses on controlling and managing the expression of those negative emotions. When the negative emotions caused by interpersonal conflict are stimulated, individuals with political skills reduce their behavior toward others [66], which can help them avoid the situation and become a negative situation [67]. In addition, in response to project uncertainty, partners with political skills may take positive action to organize the uncertain environments [63]. Partners with higher political skills actively seek opportunities to find new ways of solving uncertainties and problems [70].

Therefore, politically skilled partners regard the relationship conflicts caused by project uncertainty as opportunities rather than threats. When such individuals encounter relationship conflicts at work, they detect other people's feelings and regard those feelings as signals of important information rather than personal attacks [39]. In this way, it is possible for such individuals to form strong conflict resolutions and prevent value-related human struggles. At this point, continuing relationships are established among partners. Therefore, we can conclude that the progression of political skills is a key aspect of our understanding of project uncertainty and our ability to manage partner relationship continuity.

**H4a.** The relationship between relationship conflict and relationship continuity is weaker for partners with high levels of political skill than for those with lower levels of political skill.

**H4b.** Political skill moderates the indirect effect of project uncertainty on relationship continuity (through relationship conflict). Specifically, this indirect effect is weaker for partners with high levels of political skill than for those with lower levels of political skill.

## Methodology

### Sample and procedures

All procedures were conducted in compliance with the American Psychology Association (APA) ethics code and approved by an institute with which all authors were affiliated (Hebei GEO University), although the university did not have a formal Institutional Review Board. Notably, the study does not fall within the field of clinical psychology. In addition, all participants provided written informed consent when they were completing the questionnaires.

The sample used for this study was recruited from a set of project owners, contractors and material suppliers associated with 50 large construction projects in China. To test the above assumption, the present study used an online questionnaire survey. First, we organized a pilot test to ensure that the study methods were in line with our intentions. Five construction professionals were invited to comment on the readability, comprehensiveness and precision of the questionnaire. Their feedback was incorporated to finalize the survey instrument. Subsequently, the questionnaire was distributed to individuals linked to the 50 construction projects. The participants were required to meet the following conditions: (1) currently performing construction project work directly and (2) possess more than five years of project experience. The questionnaire was divided into two sections. Demographic and control variables part, participants need to provide basic information such as the turnover of their company, and the type of work they performed on their last project. Construct measurement part, the participants were asked to answer 31 questions related to the aims of this research. The questions included in the survey focused on the respondent's project [71], thus enabling us to determine the statistical relationships among perceptions of project uncertainty, relationship conflict and relationship continuity. To increase the response rate, follow-up emails were sent two weeks after the initial distribution. Ultimately, 230 valid responses were gathered. Ninety-seven (42%) of them were in senior management, sixty-eight (30%) were in middle management, and the remainder were working at the operational level.

To investigate nonresponse bias, the differences between respondents and nonrespondents were analyzed with regard to the size of the project team, and the number of team members. The $t$ test results ($p > 0.05$) revealed no obvious differences, thus indicating that nonresponse is not a problem to the aspects mentioned above. Moreover, the potential common method bias was checked by employing a two-step process [72]. First, a series of single factor analyses are conducted on each constructed metric, and there is no single factor that can explain the covariance. In addition, Whitney's (2001) marker variable method was applied [73]. The survey involved a question that was not associated with the topic at hand, and this question was related to the derived constructs. According to these two tests, there exist no common method bias.

### Measures

**Project uncertainty.** We measured project uncertainty using eight items. The items referred to the studies performed by Barki et al. [74], Song and Montoya-Weiss [75], Sakka et al. [76] and Jun et al. [77]. The respondents expressed the degree to which they agreed with a series of statements. An example item is "During the project, the project encountered problems that were very difficult to resolve." Responses were scored via a five-point Likert scale ranging from 1 "Strongly Disagree" to 5 "Strongly Agree". Cronbach's α was set as 0.92.

**Relationship conflict.** Drawing from previous research [9], we measured relationship conflict based on four-item scale. For example, respondents expressed whether conflicts among members were featured on the basis of a relationship-associated problem. An example item is "To what extent are personality clashes present in the project?". Responses were scored ranging from 1 "Strongly Disagree" to 5 "Strongly Agree". The relevant alpha was 0.93.

**Relationship continuity.** Relationship continuity was measured using three items, which were referred to the research performed by Celuch et al. [1]. We measured project uncertainty using three items. An example item is "We expect to continue our relationship with this partner for several years". Responses were scored ranging from 1 "Strongly Disagree" to 5 "Strongly Agree". The relevant alpha was 0.85.

**Political skill.** We used Vigoda-Gadot and Meisler's [78] shortened the eight-item version of the self-reported PSI, which itself is adapted from Ferris et al. [34]. The scale comprises four dimensions: two items were used to assess social astuteness (e.g., "I have good intuition or savvy with regarding to presenting myself to others"), two items were used to assess interpersonal impact (e.g., "I always seem to know the right thing to say or do to influence others instinctively"), two items were used to assess networking performance (e.g., "I spend a lot of time and effort at work networking with others"), and two items were used to assess apparent sincerity (e.g., "When communicating with others, I try to be genuine in what I say and do"). The respondents needed to self-report their own political skills. These items were scored ranging from 1 "Completely Disagree" to 5 "Completely Agree". The relevant alpha was 0.62.

**Control variables.** We included gender (1 = male; 2 = female), tenure (number of years), and work status (1 = full-time worker; 2 = contingent worker) as basic control variables in the analyses [79,80].

### Reliability and validity

The confirmatory factor analysis (CFAs) method was applied to check the reliability of the scales. This research applied Larcker's [81] method to determine the scale reliability. All the composite reliability values were above the threshold of 0.7 [82]. Moreover, we tested the AVE values (ranging from 0.55 to 0.92) for each scale, which mostly reached an acceptable level [82]. In addition, the square roots of the AVE values for each scale were greater than those of any of the intercorrelations among the variables. The test results were acceptable. This research used a CFA model that included project uncertainty, relationship conflict, relationship continuity and political skill. The results revealed that $\chi^2/df$ was 1.59, and the root mean square error was 0.000. Moreover, the appendix shows the factor loadings of the items. These factor loadings provide evidence to support convergent validity [81]. In the appendix, the composite reliability values for every scale are presented.

## Results and analysis

Table 1 shows the means, standard deviations, and intercorrelations for all the variables. Project uncertainty and relationship conflict were positively associated ($r = 0.31$, $p < 0.01$), whereas they were negatively associated with relationship continuity ($r = -0.26$, $p < 0.01$). Political skill also does not exhibit any obvious relationships with any other variables. Moreover, none of the control variables were significantly related to the main effect variables.

### Tests of the mediating effects

To test the proposed hypotheses, we applied the bootstrapping procedure. This procedure involves repeated random sampling and calculates the statistic for every resample. On the basis of the above results, an empirical approximation of the statistic was applied for hypothesis testing [83,84]. We employed Preacher and Hayes' [85]bootstrapping macros

**Table 1. Descriptive statistics and correlation coefficient matrix.**

| Variable | | M | SD | 1 | 2 | 3 | 4 | 5 | 6 | 7 |
|---|---|---|---|---|---|---|---|---|---|---|
| 1 | Tenure | 16.29 | 14.51 | – | | | | | | |
| 2 | Gender | – | – | 0.01 | – | | | | | |
| 3 | work status | – | – | 0.02 | 0.01 | – | | | | |
| 4 | Project uncertainty | 3.50 | 0.78 | 0.07 | 0.06 | 0.05 | – | | | |
| 5 | Relationship conflict | 2.70 | 1.03 | 0.11 | 0.08 | 0.12 | 0.31** | – | | |
| 6 | Relationship continuity | 3.20 | 0.83 | 0.01 | 0.02 | 0.05 | –0.26** | –0.49** | – | |
| 7 | Political skill | 3.41 | 0.37 | 0.19 | 0.13 | 0.07 | 0.26 | –0.12 | 0.05 | – |

**Note:** * $p < 0.05$, ** $p < 0.01$.

in SPSS software to test the mediating and moderating effects pertaining to the connections among project uncertainty, relationship conflict, relationship continuity and political skill.

Table 2 presents the results pertaining to Hypotheses 1–3. Hypothesis 1 posits that project uncertainty is positively related to relationship conflict, corresponding to an obvious standard coefficient ($B=0.31$, $t=4.84$, $p<0.01$). Hence, Hypothesis 1 was accepted. Moreover, the inverse association between relationship conflict and relationship continuity (Hypothesis 2) was supported after controlling for project uncertainty ($B=-0.45$, $t=-7.40$, $p<0.01$). Finally, in support of Hypothesis 3, project uncertainty can indirectly impact relationship continuity and is negative (–0.15). The significance test results indicated that the indirect effect was obvious (Sobel $z=-4.02$, $p<0.01$). The bootstrap results confirmed the outcomes of the Sobel test (see Table 2) with regard to the derivative effect, which did not contain zero (–0.22, –0.08). Hence, Hypotheses 1–3 were supported.

## Tests of the moderated mediation model

Subsequently, this research investigated the moderating effects of political skill on the association in relationship conflict and relationship continuity. We further examined whether this factor moderated the indirect influence of project uncertainty on relationship continuity through relationship conflict. The results shown in Table 3 indicate that the cross-product term of relationship conflict and political skill in that model (featuring relationship continuity as the dependent variable) was obvious ($B=-0.39$, $t=-8.31$, $p<0.01$). These findings are consistent with the expectations (and support Hypothesis 4a). To support Hypothesis 4a more fully, we plotted simple slopes (see Fig 2) at SD above and below the mean of the measure. It is not difficult to find that the negative association between relationship conflict and association continuity becomes weaker when greater political skill. This result supports Hypothesis 4a.

We also proved the conditional derivative effect of project uncertainty on relationship continuity (through relationship conflict). As shown in Table 3, these effects of relationship continuity were significant under all three conditions: low, mean, high ($p<0.01$). In total, this result indicates that political skill weakens the mediating effect of relationship conflict on the association with project uncertainty, which is consistent with Hypothesis 4b.

## Discussion and conclusion

This study set out to investigate the mechanisms through which project uncertainty impacts relationship continuity in construction projects, with a specific focus on the mediating role of relationship conflict and the buffering role of political skill.

**Table 2. Regression Results for Mediation.**

| Variable | | | *B* | *SE* | *t* | *p* |
|---|---|---|---|---|---|---|
| Direct and total effects | | | | | | |
| Relationship continuity regressed on project uncertainty | | | –0.26 | 0.07 | –4.12 | 0.000 |
| Relationship conflict regressed on project uncertainty | | | 0.31 | 0.08 | 4.84 | 0.000 |
| Relationship continuity regressed on relationship conflict, controlling for uncertainty | | | –0.45 | 0.05 | –7.40 | 0.000 |
| Relationship continuity regressed on project uncertainty, controlling for relationship conflict | | | –0.127 | 0.06 | –2.10 | 0.037 |
| | Value | *SE* | LL 95% CI | UL 95% CI | *z* | *p* |
| Indirect effect and significance using distribution | | | | | | |
| Sobel | –0.15 | 0.03 | –0.22 | –0.08 | –4.02 | 0.000 |
| | *M* | *SE* | LL 99% CI | UL 99% CI | | |
| Bootstrap results for indirect effect | | | | | | |
| Effect | –0.15 | 0.04 | –0.22 | –0.08 | | |

**Note:** Sample size: 230. Number of bootstrap resample=3,000. LL=lower limit; UL=upper limit; CI=confidence interval.

**Table 3. Regression Results for Conditional Indirect Effects.**

| Predictor | | B | SE | t | p |
|---|---|---|---|---|---|
| Relationship conflict | | | | | |
| **Constant** | | 1.28 | 0.30 | 4.27 | 0.000 |
| **Project uncertainty** | | 0.41 | 0.08 | 4.84 | 0.000 |
| **Relationship continuity** | | | | | |
| **Constant** | | 4.39 | 0.48 | 9.18 | 0.000 |
| **Project uncertainty** | | −0.15 | 0.07 | −2.18 | 0.031 |
| **Relationship conflict** | | −0.35 | 0.05 | −7.09 | 0.000 |
| **Political skill** | | 0.08 | 0.14 | 0.58 | 0.564 |
| **Relationship conflict × Political skill** | | −0.39 | 0.105 | −8.31 | 0.000 |
| **Moderator** | Level | Boot indirect effect | Boot SE | Boot z | Boot p |
| **Political skill** | Low | −0.24 | 0.06 | −3.15 | 0.002 |
| | Mean | −0.22 | 0.06 | −3.07 | 0.002 |
| | High | −0.22 | 0.06 | −3.05 | 0.002 |

**Note**: Sample size: 230. Number of bootstrap resample = 3,000.

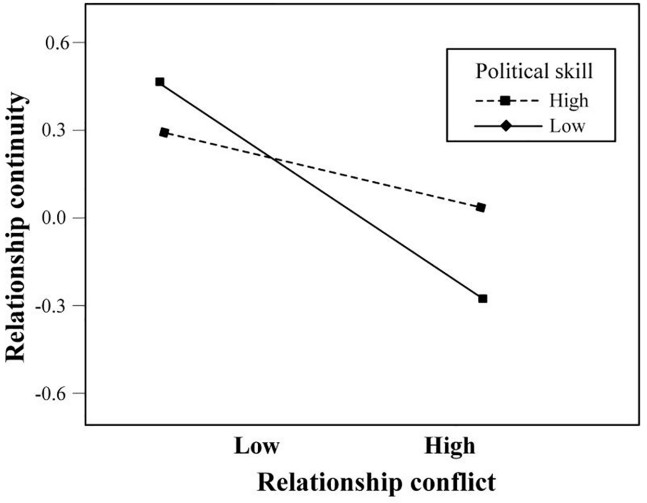

**Fig 2. Relationship continuity predicted by relationship conflict moderated by political skill.**

The results supported our hypothesized model, revealing that project uncertainty exacerbates relationship conflict, which in turn diminishes the desire for ongoing collaboration. Furthermore, the political skill of project partners significantly weakens this negative indirect effect. These findings offer nuanced insights into the social dynamics of construction project management and extend existing theories by introducing political skill as a critical contingency factor.

## Theoretical implications

Our research contributes to the theoretical understanding of relationship management in projects in several meaningful ways. First, Our research provides direct benefits to our theoretical understanding of this topic in several ways. First, by exploring the implications of project uncertainty, this research highlights the fact that uncertainty is an antecedent of

relationship conflict. Construction projects are essentially limited in terms of time, cost and resources, thus entailing that it is more difficult for participants in the project to identify the relevant goals, resources, and environment than would be the case in the context of a stable and continuous process. In construction projects, the elimination of uncertainty often resembles the struggle against Hydra (Referenced from Greek mythology, meaning a multi-headed serpent; used here metaphorically to illustrate the persistent and multiplying nature of uncertainty.), in which context cutting off any head causes two more heads to grow. Although it is possible to eliminate uncertainty in projects, this approach is not put into practice because the temporary character of such projects and the corresponding uncertainty are inevitable. In this research, the researchers focused on three types of project uncertainty: project scope, technological complexity and project novelty. Dynamic changes in the project scope often lead to the blurring of the boundaries of responsibilities. When there are deviations between the work content specified in the contract and the actual implementation process, the differences in the determination of responsibilities among different participating entities on the basis of their own interests are highly likely to trigger conflicts. Taking an EPC's (Engineering, Procurement, and Construction) general contracting project as an example, the adjustment of the construction scope caused by design changes often leads to disputes over rights and responsibilities among the design party, the construction party and the employer. The cognitive barriers brought about by technical complexity should not be underestimated either. During the application of new technologies and new processes, the cognitive differences in technical solutions among various participating entities due to differences in professional backgrounds evolve into an opposing situation in the decision-making process. The innovation of the project significantly increases the unknown risks. In the context of lacking mature experience as a reference, the differences among the project entities in risk response strategies will increase the difficulty of communication and coordination, and then lead to strained relationships. The more novel the project is for partners, the more accidents and problems they encounter. In this case, communication and interaction are viewed as important factors regarding resolving all kinds of problems caused by project uncertainty. Nevertheless, project partners are characterized by individual diversity in terms of diverse ideas and beliefs. It is difficult for project partners to coordinate and process the great amount of uncertain information they face. On the bases of the self-categorization theories, project partners are motivated to maintain their self-identity and social identity, and show a preference in favor of others who are likely to exhibit similar features to themselves. Specifically, they believe that their values can thus be reinforced. Negative feedback, such as disgust and aggressive behavior, may lead to incompatibilities in terms of different values, personalities and beliefs. As such, relationship conflicts arise. Our research highlights the claim that project uncertainty causes relationship conflicts in projects.

Second, this study verified the relationship between relationship conflict and relationship continuity. We also examined how project uncertainty influences relationship continuity through relationship conflict in the context of construction projects. Relationship conflict usually causes project partners to experience negative emotions, including anxiety, anger and stress. These negative emotions subsequently influence communication and information exchange behavior during projects. For example, when there is a conflict between the contractor and the supervision party due to quality acceptance standards, the two parties may fall into an adversarial mood, resulting in behaviors such as deliberate delays and negative responses in subsequent work, which directly disrupt the normal work collaboration process. This emotional depletion not only affects the work efficiency of individuals, but also creates a negative atmosphere at the organizational level, hindering the smooth flow of information and the sharing of knowledge. Furthermore, it weakens the willingness of stakeholders to cooperate and poses a threat to the sustainability of the relationship. Communication provides an understanding of the intentions of partners and facilitates trust and information exchange, which promote long-term partnerships. Therefore, relationship conflict influences the relationship continuity among project partners. Moreover, such projects are characterized by high levels of uncertainty. The success of a project relies on the existence of a high degree of interaction and communication. As long as interactions among project partners occur, relationship conflicts are inevitable, thus influencing relationship continuity.

Third, this research highlights that the role of political skill as an effective moderating factor may mitigate the negative influence of project uncertainty on relationship continuity. Contemporary project organizations are becoming more dynamic, flexible, and uncertain. Therefore, construction projects are political in nature. Project partners spend less time performing individual tasks; instead, cooperative interactions are common elements of successful projects. As this research indicated, political skills are viewed as important 'soft factors' for promoting relationship development and maintenance in the context of construction projects. For example, in the context of the uncertainties in the construction process triggered by the application of intelligent construction technologies, managers with high political skills can, by organizing cross-departmental seminars, guide all parties to shift their perspective from the "technical risk" angle to the "innovation and collaboration" angle. This can effectively reduce the anxiety and the tendency towards conflict caused by uncertainties, and lay a cognitive foundation for the sustainability of relationships. Moreover, political skill is a comprehensive social skill that can take cognitive, affective and behavioral forms. Politically skilled project partners exhibit social insight, interpersonal leadership, social aptitude, and sincerity. Social insight's project partners can be aware of diverse project circumstances and understand social interactions accurately, thus enabling them to address a variety of uncertain situations skillfully. Project partners who are skilled at interpersonal leadership appear to be able to understand the relevant circumstances and elicit the desired responses from others. Project partners who exhibit high levels of social aptitude can easily develop rapport with others and establish significant alliance relationships, thus rendering this characteristic highly valuable and essential regarding promoting communication and trust. Project partners who exhibit sincerity tend to elicit perceptions of integrity and genuineness from others, thus helping them eliminate conflict. Overall, politically skilled project partners can read uncertain project situations and personal interactions, utilize their skill at influencing people in a sincere way and obtain the desired outcomes from other partners. Project partners who exhibit a high degree of political skill can weaken the dysfunctional effect of relationship conflict by controlling for negative emotions. In the face of project uncertainty, politically skilled project partners regard the relationship conflicts caused by project uncertainty as opportunities rather than threats. They seek opportunities and important information that can enable them to identify new ways of resolving uncertainties and problems rather than viewing them as personal attacks thus enabling them to avoid meaningless human struggle. Naturally, trust, a win-win mentality and continuous relationships are established among project partners. Accordingly, this research extends the theory of political skill by identifying it as an important moderator with in the context of construction project management.

## Managerial implications

The findings also offer insights that are relevant to project managers or stakeholders. A construction project is known to constitute an uncertain work environment because of the great variety of external and uncontrolled variables that influence the achievement of project objectives. Moreover, project partners exhibit individual diversity in terms of their characteristics and have different agendas and backgrounds. Thus, relationship conflict is inevitable as long as interactions occur among partners throughout the project. In addition, project managers should attach importance to the task of maintaining long-term partner relationships. Project managers should recognize the objective facts listed above, which are prevalent in contemporary projects. Previous studies have investigated the formal project management approaches used to achieve project objectives in detail.

This research highlights the role of flexible factors in project management. As the research shows, project activities are characterized by political attributes. Project partners' political skill is an essential soft skill in daily project work. Politically skilled project partners view the relationship conflicts caused by project uncertainty as opportunities and seek to take a positive approach to the task of resolving uncertain problems. In this way, project partners develop practical strategies such as fostering open communication, establishing clearer contractual terms, or implementing effective conflict resolution frameworks, thus enabling them to avoid meaningless human struggle. Accordingly, continuing relationships are established among partners.

Finally, some suggestions have been made to improve and enhance individuals' political skills in project management. For instance, project human resources and organizational development departments should view political skill as an important person ability that is related to the recruitment process as well as promotion and project team building. On the one hand, the project human resources department could evaluate employees who are to be recruited and promoted using measurements of political skill. Candidates who exhibit a high level of political skill should be preferred. Moreover, project organizational development departments can choose politically skilled individuals to serve as project team members to enhance the project's overall executive capability at the team construction stage. On the other hand, some political skill training programs should be designed and developed for all project participants. Ferris et al. (2005) analyzed a variety of ways in which team members can be trained to enhance their political skills, such as through drama-based training, behavior modeling, and criticism sessions [34]. For example, through case analysis and role-playing, the participating parties can be enabled to learn how to effectively communicate and coordinate with different stakeholders in a complex project environment to enhance their political acumen and response capabilities. In addition, arrange for the participating parties should be responsible for handling some sensitive issues in the project or for communicating and coordinating with difficult stakeholders. By solving these practical problems, they can continuously summarize experiences, improve their methods, and enhance their ability to apply political skills in complex environments. In summary, when a project includes many politically skilled members, it produces more positive results [34]. Political skill is a secret weapon that project managers can use to manipulate relationships with other partners to ensure project success.

## Limitations and further research

This research has the following limitations. First, related information and data were largely from the construction industry. In the future, it is necessary to check other industries for a wider applicability. Second, political skill was only reflected only via self-reported data. Therefore, further research can incorporate peer-peer reports to improve the reliability of research conclusions.

## Supporting information

**S1 File. Data.**
(XLS)

## Acknowledgments

This study was supported by National pre-research project of Hebei GEO University (KY2024YB18); Hebei Technology Innovation Center for Intelligent Development, Control of Underground Built Environment; Hebei International Underground Space Associated Research Centers;  Demonstration Undergraduate Majors for Applied-Oriented Transformation in Hebei Province.

## Author contributions

**Conceptualization:** Xiaoyan Huo.

**Formal analysis:** Xiaoyan Huo.

**Investigation:** Xiaoyan Huo, Shiya Gao, Huiyang Zhang.

**Methodology:** Shiya Gao, Huiyang Zhang.

**Project administration:** Xiaoyan Huo.

**Resources:** Xiaoyan Huo.

**Writing – original draft:** Xiaoyan Huo.

**Writing – review & editing:** Shiya Gao, Huiyang Zhang.

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
