## [Decision Letter · Decision Letter 0]

11 Dec 2024

Dear Dr. Huo,

Thank you for submitting your manuscript to PLOS ONE. After careful consideration, we feel that it has merit but does not fully meet PLOS ONE’s publication criteria as it currently stands. Therefore, we invite you to submit a revised version of the manuscript that addresses the points raised during the review process.

We look forward to receiving your revised manuscript.

Kind regards,

Tachia Chin

Academic Editor

PLOS ONE

Journal Requirements:

4. In the online submission form you indicate that your data is not available for proprietary reasons and have provided a contact point for accessing this data. Please note that your current contact point is a co-author on this manuscript. According to our Data Policy, the contact point must not be an author on the manuscript and must be an institutional contact, ideally not an individual. Please revise your data statement to a non-author institutional point of contact, such as a data access or ethics committee, and send this to us via return email. Please also include contact information for the third party organization, and please include the full citation of where the data can be found.

Reviewers' comments:

Reviewer's Responses to Questions

**Comments to the Author**

1. Is the manuscript technically sound, and do the data support the conclusions?

Reviewer #1: Partly

Reviewer #2: Yes

2. Has the statistical analysis been performed appropriately and rigorously?

Reviewer #1: Yes

Reviewer #2: I Don't Know

3. Have the authors made all data underlying the findings in their manuscript fully available?

Reviewer #1: Yes

Reviewer #2: Yes

4. Is the manuscript presented in an intelligible fashion and written in standard English?

Reviewer #1: Yes

Reviewer #2: Yes

Reviewer #1: This paper investigates how project uncertainty and relationship conflict influence the continuity of relationships in the construction industry. Given the complex, dynamic nature of construction projects, understanding how these factors affect long-term collaborations is crucial for enhancing project success and fostering sustainable relationships between stakeholders.

One of the main weaknesses of the paper is the lack of a strong, detailed conceptual framework. While the relationship between project uncertainty, conflict, and relationship continuity is established, the underlying theories and mechanisms that connect these concepts are not fully explored. A more robust theoretical grounding could help clarify how project uncertainty and conflict lead to disruptions in relationships and why these disruptions occur in certain contexts but not others.

he paper provides valuable insights into the theoretical aspects of project uncertainty and relationship conflict, but it falls short in offering practical recommendations for construction managers or stakeholders. For example, practical strategies like fostering open communication, establishing clearer contractual terms, or implementing conflict resolution frameworks would significantly enhance the paper’s applicability to real-world situations.

Reviewer #2: The paper entitled” The impacts of project uncertainty and relationship conflict on relationship continuity in construction: The role of political skill” deals with a very interesting topic. In general, I appreciate the aims of this work; it is quite interesting and informative to most readers of this field.

However, I have the following comments that hopefully help the authors improve their paper:

• Regarding the literature review, I strongly recommend that the authors include a summary table of their comprehensive literature review. Such a table would not only help identify gaps in the existing literature but also enhance the contribution of this work by providing a clear overview of the relevant research in the field.

• As usual, thorough proof-reading is recommended.

I wish the author(s) all the best for their research and that these comments will be useful to them in improving the paper.

**Do you want your identity to be public for this peer review?** For information about this choice, including consent withdrawal, please see our Privacy Policy

Reviewer #1: No

Reviewer #2: No

---

## [Author Response · Author response to Decision Letter 1]

18 Jan 2025

Dear editors and reviewers:

Thank you for reviewer’s comments and suggestions which are relevant for us. Those comments are all valuable and very helpful for revising and improving our paper, as well as an important guiding to our research. According to the comments, we have made a significant revision. The changes have been highlighted within the 'Revised Manuscript with Track Changes' by using the track changes mode in MS Word. We also upload a separate file labeled 'Manuscript'.

The comments and responses as follows:

Reviewer #1:

This paper investigates how project uncertainty and relationship conflict influence the continuity of relationships in the construction industry. Given the complex, dynamic nature of construction projects, understanding how these factors affect long-term collaborations is crucial for enhancing project success and fostering sustainable relationships between stakeholders.

1. One of the main weaknesses of the paper is the lack of a strong, detailed conceptual framework. While the relationship between project uncertainty, conflict, and relationship continuity is established, the underlying theories and mechanisms that connect these concepts are not fully explored. A more robust theoretical grounding could help clarify how project uncertainty and conflict lead to disruptions in relationships and why these disruptions occur in certain contexts but not others.

Response: Thank you for reviewers careful review. According to reviewer’s good suggestion, we have carefully revised the paper. Firstly, in order to make the conceptual framework clearer, based on transaction cost theory Line 153��self-classification theory Line 171�, social cognition theory Line 197� and contingency theory Line 243�, the paper reorganizes and analyzes the variables and the relationship between variables. Secondly, the manuscript rearranges the research progress, increases the research gap and the problems to be solved in this research. Especially in the last paragraph of the introduction, the problems to be solved in this study are clarified one by one.

2. The paper provides valuable insights into the theoretical aspects of project uncertainty and relationship conflict, but it falls short in offering practical recommendations for construction managers or stakeholders. For example, practical strategies like fostering open communication, establishing clearer contractual terms, or implementing conflict resolution frameworks would significantly enhance the paper’s applicability to real-world situations.

Response: Thank you for reviewer’s good suggestion. Through the research of this thesis, project managers should have a correct understanding of project uncertainties and realize that project uncertainties can affect the long - term cooperative relationships among stakeholders through relationship conflicts. Previous studies have introduced many formal governance mechanisms for maintaining the cooperation of project stakeholders (such as contractual constraints, regular meetings, etc.). This research proposes the role of flexible factors in project cooperation. Politically skilled project partners view the relationship conflicts caused by project uncertainty as opportunities and seek to take a positive approach to the task of resolving uncertain problems. Some suggestions have been made to improve and enhance individuals’ political skills in project management. For instance, project human resources and organizational development departments should view political skill as an important person ability that is related to the recruitment process as well as promotion and project team building. On the other hand, some political skill training programs should be designed and developed for all project participants. It is believed that these practical suggestions can serve as a reference for project managers in their project management.

Reviewer #2: 

The paper entitled” The impacts of project uncertainty and relationship conflict on relationship continuity in construction: The role of political skill” deals with a very interesting topic. In general, I appreciate the aims of this work; it is quite interesting and informative to most readers of this field. However, I have the following comments that hopefully help the authors improve their paper:

1. Regarding the literature review, I strongly recommend that the authors include a summary table of their comprehensive literature review. Such a table would not only help identify gaps in the existing literature but also enhance the contribution of this work by providing a clear overview of the relevant research in the field.

Response: Thank you for reviewers careful review. As the reviewers said, listing a literature review table does have a very good effect on classifying the literature of this research and providing a clear overview of the relevant research in the field. Nevertheless, in order to present the logical relationships and hypotheses between variables one by one, numerous literatures are repeatedly cited in the literature review section. Therefore, this paper does not list the references in summary table. secondly, in a previous study by my team members, a detailed literature review of “project uncertainty” was conducted and presented in table form (Antecedents of relationship conflict in cross-functional project teams. Project Management Journal, 2016, 47(5): 52-69.). Thirdly, in order to make the conceptual framework clearer, based on transaction cost theory Line 153��self-classification theory Line 171�, social cognition theory Line 197� and contingency theory Line 243�, the paper reorganizes and analyzes the relationship between variables. It is hoped that through the above - mentioned revisions, the paper can present a clear overview of the relevant research in this field.

2. As usual, thorough proof-reading is recommended. I wish the author(s) all the best for their research and that these comments will be useful to them in improving the paper.

Response: According to the reviewers' suggestions, this paper has gone through the proof - reading process. This paper has improved in terms of expression and readability.

---

## [Decision Letter · Decision Letter 1]

21 Mar 2025

Dear Dr. Huo,

Thank you for submitting your manuscript to PLOS ONE. After careful consideration, we feel that it has merit but does not fully meet PLOS ONE’s publication criteria as it currently stands. Therefore, we invite you to submit a revised version of the manuscript that addresses the points raised during the review process.

We look forward to receiving your revised manuscript.

Kind regards,

Tachia Chin

Academic Editor

PLOS ONE

Journal Requirements:

Additional Editor Comments :

1. First and foremost: The linguistic accuracy and professional tone of the manuscript. To ensure that your research conveys its findings with the utmost clarity and professionalism, I sincerely recommend that the manuscript undergoes copyediting by a specialized agency.

2. The discussion and conclusion section needs to be more nuanced and analytical. The authors should critically evaluate their findings, comparing them with existing literature and addressing any discrepancies or inconsistencies.

Reviewers' comments:

Reviewer's Responses to Questions

**Comments to the Author**

Reviewer #1: All comments have been addressed

Reviewer #2: All comments have been addressed

2. Is the manuscript technically sound, and do the data support the conclusions?

Reviewer #1: Yes

Reviewer #2: Yes

3. Has the statistical analysis been performed appropriately and rigorously?

Reviewer #1: Yes

Reviewer #2: I Don't Know

4. Have the authors made all data underlying the findings in their manuscript fully available?

Reviewer #1: Yes

Reviewer #2: (No Response)

5. Is the manuscript presented in an intelligible fashion and written in standard English?

Reviewer #1: Yes

Reviewer #2: Yes

Reviewer #1: (No Response)

Reviewer #2: The manuscript has significantly improved as compared to the previous version. Indeed, the authors tried to improve it, and the main weaknesses are solved.

**Do you want your identity to be public for this peer review?** For information about this choice, including consent withdrawal, please see our Privacy Policy

Reviewer #1: No

Reviewer #2: No

---

## [Author Response · Author response to Decision Letter 2]

7 May 2025

Dear editors and reviewers:

Thank you for reviewer’s comments and suggestions which are relevant for us. Those comments are all valuable and very helpful for revising and improving our paper, as well as an important guiding to our research. According to the comments, we have made a significant revision. The changes have been highlighted within the ‘Revised Manuscript with Track Changes’ by using the track changes mode in MS Word. We also upload a separate file labeled ‘Manuscript’.

The comments and responses as follows:

Response:

We have reviewed the reference list to ensure that it is complete and correct. The reference list have not cited papers that have been retracted.

2. First and foremost: The linguistic accuracy and professional tone of the manuscript. To ensure that your research conveys its findings with the utmost clarity and professionalism, I sincerely recommend that the manuscript undergoes copyediting by a specialized agency.

Response:

We accepted editors suggestions regarding the language problem. We have already had the entire text linguistically polished by a professional institution (AJE) to ensure that our research conveys its findings with the utmost clarity and professionalism.

3. The discussion and conclusion section needs to be more nuanced and analytical. The authors should critically evaluate their findings, comparing them with existing literature and addressing any discrepancies or inconsistencies.

Response:

We fully agree with this suggestion. Following the suggestion, we have added a large amount of analysis to the conclusion section. The specific revisions can be seen in the “Revised Manuscript with Track Changes”.

---

## [Decision Letter · Decision Letter 2]

18 Aug 2025

Dear Dr. Huo,

Thank you for submitting your manuscript to PLOS ONE. After careful consideration, we feel that it has merit but does not fully meet PLOS ONE’s publication criteria as it currently stands. Therefore, we invite you to submit a revised version of the manuscript that addresses the points raised during the review process.

We look forward to receiving your revised manuscript.

Kind regards,

Tachia Chin

Academic Editor

PLOS ONE

Journal Requirements:

Additional Editor Comments :

Carefully consider the suggestions of all reviewers and revise again.

Reviewers' comments:

Reviewer's Responses to Questions

**Comments to the Author**

Reviewer #3: All comments have been addressed

Reviewer #4: (No Response)

Reviewer #5: (No Response)

2. Is the manuscript technically sound, and do the data support the conclusions?

Reviewer #3: Yes

Reviewer #4: No

Reviewer #5: Partly

3. Has the statistical analysis been performed appropriately and rigorously?

Reviewer #3: Yes

Reviewer #4: No

Reviewer #5: Yes

4. Have the authors made all data underlying the findings in their manuscript fully available?

Reviewer #3: Yes

Reviewer #4: No

Reviewer #5: Yes

5. Is the manuscript presented in an intelligible fashion and written in standard English?

Reviewer #3: Yes

Reviewer #4: No

Reviewer #5: No

Reviewer #3: Accept the manuscript. All the reviews were addressed in the former revision. I with good luck with your research.

Reviewer #4: The manuscript titled “The impacts of project uncertainty and relationship conflict on relationship continuity in construction: The role of political skill” presents an investigation into the relational dynamics within construction projects, focusing on project uncertainty and political skill. While the topic is timely and potentially valuable for the construction management field, the paper suffers from several foundational and structural deficiencies that substantially undermine its academic rigor and suitability for publication in its current form.

One of the most critical issues in this study is the lack of a clearly established theoretical foundation. Although the manuscript briefly references the contingency theory in relation to political skill, it fails to adopt any managerial or organizational theory as a grounding framework for its conceptual model or hypotheses. This omission severely limits the paper's ability to demonstrate theoretical contribution, and the study appears to be constructed without anchoring its variables or assumptions in any established theoretical discourse. The hypotheses are presented in isolation, disconnected from a guiding framework that would validate their inclusion and enhance their interpretive power.

The abstract is another area of major concern. It is poorly written and unstructured, lacking the conventional components expected in a scholarly summary. It fails to articulate the study’s aim, theoretical grounding, methodology, major findings, and implications in a coherent sequence. The use of informal language, including the repetitive use of "we," further detracts from its academic tone. Additionally, the abstract’s formatting is inconsistent and reflects a lack of attention to publication standards, which compromises the professionalism of the presentation.

Methodologically, the paper exhibits weaknesses that raise significant questions about the validity and generalizability of the findings. The sample size of 230 participants is inadequately justified, particularly given the context of the expansive Chinese construction industry. There is no explanation of the overall population size or statistical rationale for selecting this number of respondents. Moreover, essential contextual details—such as the specific cities where the data was collected, the type of construction sectors surveyed, and the rationale for choosing these projects—are conspicuously absent. This lack of clarity obstructs the reader's ability to understand the representativeness of the sample and the relevance of the findings.

The manuscript also glosses over potential bias issues. While there is a brief mention of common method bias and two techniques are cited for its evaluation, the discussion is vague and lacks methodological depth. The treatment of common method variance is presented as a checklist rather than a critical analysis, which weakens the study’s internal validity. Furthermore, there is no substantial elaboration on how the data collection procedures addressed potential threats related to self-reported survey designs.

Equally problematic is the discussion section, which falls short of providing critical engagement with existing literature. The authors do not compare their findings with recent or relevant studies, nor do they contextualize their contributions within broader research debates in project management or organizational behavior. The absence of such comparative discussion limits the scholarly impact of the paper and hinders readers from appreciating its novelty or practical significance. The results are presented descriptively, but without analytical depth or engagement with recent empirical evidence from similar settings.

In terms of academic writing quality, the overuse of first-person pronouns and informal phrasing detracts from the professionalism of the paper. Although the manuscript claims to have undergone linguistic editing, grammatical errors and syntactical issues remain prevalent throughout the text. These issues affect the clarity and precision of the arguments, making it difficult to follow the line of reasoning in several key sections.

Overall, the manuscript exhibits substantial theoretical, methodological, and structural deficiencies. Its lack of a guiding theoretical lens, weak methodological justification, incomplete contextual information, insufficient analytical discussion, and problematic academic tone collectively render it unsuitable for publication in its current form. These issues are not minor or isolated, but systemic across the paper, suggesting the need for significant redevelopment to meet scholarly standards.

Reviewer #5: Thank you for the opportunity to review the revised manuscript on the relationship between project uncertainty and relationship continuity mediated by relationship conflict and moderated by political skills.

While the authors have made improvements based on the reviewers' suggestions, several key areas still require attention to make the manuscript publishable:

1. The introduction needs to more effectively establish the current state of research and the study's unique contribution. Additionally, the introduction of political skills should be more nuanced, acknowledging its potential negative impacts ("double-edged sword" nature) rather than presenting it solely in a positive light.

2. A citation for transaction cost theory is missing.

3. Some citations (e.g., lines 72, 75) need to be corrected. An error is also noted on line 194.

4. Lines 380-384 contain confusing information regarding the variables and instruments used for data collection.

5. The discussion section is primarily descriptive and disconnected from the existing literature, with the exception of two citations. It should be expanded to more thoroughly connect the findings to previous studies rather than relying on intuition.

6. The manuscript still contains numerous typos and awkward sentences (e.g., lines 66, 111, etc.). In Table 2, "Soble" should be corrected to "Sobel".

7. The meaning of abbreviations like EPC or Hydra must be defined for the reader.

8. The tables are also difficult to interpret and follow.

**Do you want your identity to be public for this peer review?** For information about this choice, including consent withdrawal, please see our Privacy Policy

Reviewer #3: **Yes: ** Ana junça Silva

Reviewer #4: No

Reviewer #5: No

---

## [Author Response · Author response to Decision Letter 3]

22 Sep 2025

Dear reviewers:

Thank you for reviewer’s comments and suggestions which are relevant for us. Those comments are all valuable and very helpful for revising and improving our paper, as well as an important guiding to our research. According to the comments, we have made a significant revision. The changes have been highlighted within the “Revised Manuscript with Track Changes”. We also upload a separate file labeled “Manuscript”.

The comments and responses as follows:

Reviewer #3: Accept the manuscript. All the reviews were addressed in the former revision. I with good luck with your research.

Response: I sincerely thank you for thoroughly reviewing my manuscript and recommending its acceptance.

Reviewer #4: The manuscript titled “The impacts of project uncertainty and relationship conflict on relationship continuity in construction: The role of political skill” presents an investigation into the relational dynamics within construction projects, focusing on project uncertainty and political skill. While the topic is timely and potentially valuable for the construction management field, the paper suffers from several foundational and structural deficiencies that substantially undermine its academic rigor and suitability for publication in its current form.

One of the most critical issues in this study is the lack of a clearly established theoretical foundation. Although the manuscript briefly references the contingency theory in relation to political skill, it fails to adopt any managerial or organizational theory as a grounding framework for its conceptual model or hypotheses. This omission severely limits the paper's ability to demonstrate theoretical contribution, and the study appears to be constructed without anchoring its variables or assumptions in any established theoretical discourse. The hypotheses are presented in isolation, disconnected from a guiding framework that would validate their inclusion and enhance their interpretive power.

Response: We sincerely thank the reviewer for this profound and constructive feedback. A clear theoretical foundation is crucial for enhancing the explanatory power and academic value of the research. In response to this valuable suggestion, we have refined the revised manuscript by explicitly adopting and elaborating on the “Contingency Theory” as the core theoretical framework of our entire study. (1) Strengthening the theoretical framework statement in the introduction: We have added a new paragraph in the introduction stating that this study is grounded in Contingency Theory. We pointed out that “Based on the contingency theory, there is no universally applicable optimal management method for organizations; the effectiveness of a management approach depends on the degree of alignment between management practices and the external environment (such as uncertainty). Guided by this principle, this research focuses on political skill as a contingency factor to examine the ways in which it mitigates the negative impact of project uncertainty on partner relationship continuity.” (Lines 112-117) This provides a clear theoretical perspective for the entire study. (2)In the literature review and hypothesis development section, we introduce political skill, defining it as a crucial “contingency-coping mechanism”. We elaborate on how individuals with high political skill leverage their social insight and interpersonal influence to effectively manage interactions, mediate disagreements, and rebuild trust in highly uncertain and conflict-ridden environments, thereby mitigating the negative impacts of uncertainty(Lines 251-253). Additionally, in the hypothesis development section, we also employed Transaction Cost Theory (Line 159), Information Processing Theory (Line 173), and Self-Categorization Theory (Line 177). This significantly strengthened the theoretical logic and persuasiveness of our hypothesis formulation. (3) In the discussion section, we closely aligned our findings with Contingency Theory to explain their theoretical implications: When discussing the research results, we particularly emphasized interpreting their theoretical significance from the perspective of Contingency Theory. Our findings not only validate the relationships between variables but, more importantly, reveal how “political skill” as a crucial individual trait, achieves “Person-Environment Fit” to address uncertainty as an environmental contingency factor. This significantly enhances the theoretical depth and contribution of our research conclusions.

Through these revisions, we hope the reviewers will recognize that we have successfully deeply integrated contingency theory as the “backbone” of our research, providing a much-needed theoretical anchor for the conceptual model and hypotheses. We believe the theoretical rigor and contribution of the revised manuscript have been substantially enhanced.

The abstract is another area of major concern. It is poorly written and unstructured, lacking the conventional components expected in a scholarly summary. It fails to articulate the study’s aim, theoretical grounding, methodology, major findings, and implications in a coherent sequence. The use of informal language, including the repetitive use of "we," further detracts from its academic tone. Additionally, the abstract’s formatting is inconsistent and reflects a lack of attention to publication standards, which compromises the professionalism of the presentation.

Response: We sincerely thank the reviewers for their constructive feedback regarding the abstract. We acknowledge that the previous version was inadequately structured and written in an informal tone, which failed to meet academic standards. We will thoroughly revise the abstract to include all conventional components: clearly stating the research objectives, theoretical foundation, methodology, key findings, and implications. In addition, we will ensure a coherent flow of ideas, replace informal expressions with appropriate academic language, and eliminate repetitive use of first-person pronouns. Formatting issues will be corrected in accordance with target journal guidelines. We appreciate these helpful comments and believe the revised abstract will significantly improve in clarity and professionalism.

Methodologically, the paper exhibits weaknesses that raise significant questions about the validity and generalizability of the findings. The sample size of 230 participants is inadequately justified, particularly given the context of the expansive Chinese construction industry. There is no explanation of the overall population size or statistical rationale for selecting this number of respondents. Moreover, essential contextual details-such as the specific cities where the data was collected, the type of construction sectors surveyed, and the rationale for choosing these projects-are conspicuously absent. This lack of clarity obstructs the reader's ability to understand the representativeness of the sample and the relevance of the findings.

Response: We sincerely thank the reviewer for raising these important methodological points. We agree that greater clarity and justification are necessary regarding the sample and data collection context. In response to these concerns, we will provide the following revisions in the revised manuscript: (1) We will include a more thorough justification for the sample size of 230, with reference to similar studies in the field and power analysis where applicable, to clarify its adequacy for the analyses performed. (2) We will specify the overall target population size and sampling framework (in Line 349-351) to enhance transparency regarding sample selection and representatives. (3) The criteria for selecting specific projects and participants are definitive (in Line 339-340). This will help readers better assess the generalization and relevance of the findings.

The manuscript also glosses over potential bias issues. While there is a brief mention of common method bias and two techniques are cited for its evaluation, the discussion is vague and lacks methodological depth. The treatment of common method variance is presented as a checklist rather than a critical analysis, which weakens the study’s internal validity. Furthermore, there is no substantial elaboration on how the data collection procedures addressed potential threats related to self-reported survey designs.

Response: We sincerely thank the reviewers for their constructive feedback regarding the common method variance. Regarding common method bias (CMB), a two-step procedure was adopted to test for potential common method bias. First, a series of single factor analyses are conducted on each constructed metric, and there is no single factor that can explain the covariance. In addition, Whitney’s (2001) marker variable method was applied. These validation procedures are also commonly used in academic research. We believe these revisions will significantly improve the methodological rigor of our paper and provide readers with a transparent assessment of potential biases.

Equally problematic is the discussion section, which falls short of providing critical engagement with existing literature. The authors do not compare their findings with recent or relevant studies, nor do they contextualize their contributions within broader research debates in project management or organizational behavior. The absence of such comparative discussion limits the scholarly impact of the paper and hinders readers from appreciating its novelty or practical significance. The results are presented descriptively, but without analytical depth or engagement with recent empirical evidence from similar settings.

Response: We sincerely thank the reviewer for this critical feedback regarding the discussion section. We acknowledge that our original discussion lacked sufficient comparative analysis with recent literature and deeper engagement with broader scholarly debates, which limited the articulation of our study’s contributions. In response to this comment, we will thoroughly revise the discussion section to achieve the following: (1) We will contextualize the contributions of our results within ongoing research debates-particularly those related to political skill, and uncertainty in construction projects-to better highlight the theoretical and practical significance of our work. (2) We will strengthen the analytical depth of the discussion to interpret their implications in light of existing empirical and theoretical literature. We believe these revisions will significantly enhance the scholarly impact of the paper and allow readers to more fully appreciate the novelty and relevance of our findings.

In terms of academic writing quality, the overuse of first-person pronouns and informal phrasing detracts from the professionalism of the paper. Although the manuscript claims to have undergone linguistic editing, grammatical errors and syntactical issues remain prevalent throughout the text. These issues affect the clarity and precision of the arguments, making it difficult to follow the line of reasoning in several key sections.

Response: We thank the reviewer for highlighting these important issues concerning the academic writing quality of our manuscript. We sincerely apologize for the shortcomings in language and style, which have impacted the clarity and professionalism of the paper. In response to this feedback, we will undertake a comprehensive revision of the entire manuscript to address these concerns. Specifically, we will: (1) Significantly reduce the use of first-person pronouns and replace them with more formal and objective academic phrasing where appropriate. (2) Carefully review and correct grammatical errors, syntactical inaccuracies, and informal expressions throughout the text. (3) Ensure that sentence structures are clear and precise to strengthen the logical flow and readability of the arguments. We will also seek additional professional language editing services to ensure the revised manuscript meets the high linguistic standards expected for publication. We appreciate the reviewer’s careful reading and constructive suggestions, which will greatly improve the overall quality of our paper.

Overall, the manuscript exhibits substantial theoretical, methodological, and structural deficiencies. Its lack of a guiding theoretical lens, weak methodological justification, incomplete contextual information, insufficient analytical discussion, and problematic academic tone collectively render it unsuitable for publication in its current form. These issues are not minor or isolated, but systemic across the paper, suggesting the need for significant redevelopment to meet scholarly standards.

Response: We are dedicated to addressing each of the points raised above and believe that these revisions will significantly improve the quality and suitability of the manuscript for publication. We greatly value the opportunity to revise and resubmit a substantially improved version and thank the reviewers again for their invaluable guidance.

Reviewer #5: Thank you for the opportunity to review the revised manuscript on the relationship between project uncertainty and relationship continuity mediated by relationship conflict and moderated by political skills.

While the authors have made improvements based on the reviewers' suggestions, several key areas still require attention to make the manuscript publishable:

1. The introduction needs to more effectively establish the current state of research and the study's unique contribution. Additionally, the introduction of political skills should be more nuanced, acknowledging its potential negative impacts ("double-edged sword" nature) rather than presenting it solely in a positive light.

Response: We sincerely thank you for this insightful suggestion. We have revised the Introduction section to better contextualize the current research landscape and more clearly articulate the unique contributions of our study. Additionally, we have incorporated a more balanced view of political skill, acknowledging its potential negative implications where relevant. Please see lines 83-87 and 102-105 in the revised manuscript.

2. A citation for transaction cost theory is missing.

Response: We apologize for this oversight. We have now included a citation for Transaction Cost Theory (Williamson, 1985) in the section where it is first mentioned. Please see line 156.

3. Some citations (e.g., lines 72, 75) need to be corrected. An error is also noted on line 194.

Response: Thank you for pointing this out. We have carefully reviewed and corrected all citations throughout the manuscript.

4. Lines 380-384 contain confusing information regarding the variables and instruments used for data collection.

Response: We have rewritten this section to clarify the data collection process and the variables measured. We now more clearly distinguish between the constructs and their operationalization. Please see the revised Methodology section.

5. The discussion section is primarily descriptive and disconnected from the existing literature, with the exception of two citations. It should be expanded to more thoroughly connect the findings to previous studies rather than relying on intuition.

Response: We agree completely. We have substantially revised the Discussion section to integrate more references to prior literature and to better contextualize our findings within existing theoretical frameworks. We now explicitly compare and contrast our results with those of previous studies. Please see the revised Discussion.

6. The manuscript still contains numerous typos and awkward sentences (e.g., lines 66, 111, etc.). In Table 2, "Soble" should be corrected to "Sobel".

Response: We have thoroughly proofread the manuscript to correct typos and improve sentence fluency. “Soble” in Table 2 has been corrected to “Sobel.” We also thank you for pointing out specific lines—these have been addressed.

7. The meaning of abbreviations like EPC or Hydra must be defined for the reader.

Response: We have now defined all abbreviations upon first use. For example: EPC (Engineering, Procurement, and Construction) (Line 498); Hydra: Referenced from Greek mythology, meaning a m

---

## [Decision Letter · Decision Letter 3]

29 Sep 2025

The impacts of project uncertainty and relationship conflict on relationship continuity in construction: The role of political skill

PONE-D-24-43748R3

Dear Dr. Xiaoyan Huo,

We’re pleased to inform you that your manuscript has been judged scientifically suitable for publication and will be formally accepted for publication once it meets all outstanding technical requirements.

Kind regards,

Tachia Chin

Academic Editor

PLOS ONE

Additional Editor Comments (optional):

Reviewers' comments:

Reviewer's Responses to Questions

**Comments to the Author**

Reviewer #3: All comments have been addressed

2. Is the manuscript technically sound, and do the data support the conclusions?

Reviewer #3: Yes

3. Has the statistical analysis been performed appropriately and rigorously?

Reviewer #3: Yes

4. Have the authors made all data underlying the findings in their manuscript fully available?

Reviewer #3: Yes

5. Is the manuscript presented in an intelligible fashion and written in standard English?

Reviewer #3: Yes

Reviewer #3: The manuscript improved in its quality as the authors addressed all the comments. Thus, I recommend that it can be accepted in its present form.

**Do you want your identity to be public for this peer review?** For information about this choice, including consent withdrawal, please see our Privacy Policy

Reviewer #3: No

---

## [Editor Report · Acceptance letter]

PONE-D-24-43748R3

PLOS ONE

Dear Dr. Huo,

I'm pleased to inform you that your manuscript has been deemed suitable for publication in PLOS ONE. Congratulations! Your manuscript is now being handed over to our production team.

Kind regards,

on behalf of

Dr. Tachia Chin

Academic Editor

PLOS ONE